# VALUE FUNCTION ESTIMATION USING CONDITIONAL DIFFUSION MODELS FOR CONTROL

## ABSTRACT

A fairly reliable trend in offline reinforcement and imitation learning is that performance scales with the number of parameters, provided a complimentary scaling in amount of training data. As the appetite for large models increases, it is imperative to address, sooner than later, the potential problem of running out of high-quality demonstrations. In this case, instead of collecting only new data via costly human demonstrations, risking a simulation-to-real transfer with uncertain effects or running expensive online interactions, it would be beneficial to leverage vast amounts of readily-available low-quality data. Since classical control algorithms such as behavior cloning or temporal difference learning cannot be used on reward-free or action-free data out-of-the-box, this solution warrants novel training paradigms for continuous control. We propose a simple algorithm called Diffused Value Function (DVF), which learns a joint multi-step model of the environment-agent interaction dynamics using a diffusion model. This model can be efficiently learned from state sequences (i.e., without access to reward functions nor actions), and subsequently used to estimate the value of each action out-of-the-box. We show how DVF can be used to efficiently capture the state visitation measure of a policy, and show promising qualitative and quantitative results on challenging robotics benchmarks.

## 1 INTRODUCTION

The success of foundation models (Chowdhery et al., 2022; Touvron et al., 2023) is often attributed to their size (Kaplan et al., 2020) and abundant training data, a handful of which is usually annotated by a preference model trained on human feedback (Ouyang et al., 2022). Similarly, the reinforcement learning community, in particular robotics, has seen a surge in large multimodal learners (Brohan et al., 2023; Stone et al., 2023; Driess et al., 2023), which also require vast amounts of high-quality training demonstrations. What can we do when annotating demonstrations is prohibitively costly, and the simulation-to-real gap is too large? Recent works show that partially pre-training the agent on large amounts of low-returns data with missing information can help accelerate learning from optimal demonstration (Baker et al., 2022; Fan et al., 2022). A major drawback of these works lies in the compounding prediction error: training a preference model on optimal demonstrations and subsequently using this model in reinforcement learning (RL) or behavior cloning (BC) approaches includes both the uncertainty from the preference bootstrapping, as well as the RL algorithm itself. Instead, we opt for a different path: decompose the value function, a fundamental quantity for continuous control, into components that depend only on states, only on rewards, and only on actions. These individual pieces can then be trained separately on different subsets of available data, and re-combined together to construct a value function estimate, as shown in later sections.

Factorizing the value function into dynamics, decision and reward components poses a major challenge, since it requires disentangling the non-stationarity induced by the controller from that of the dynamical system. Model-based approaches address this problem by learning a differentiable transition model of the dynamic system, through which the information from the controller can be propagated (Yu et al., 2020; Argenson & Dulac-Arnold, 2020; Kidambi et al., 2020; Yu et al., 2021). While these approaches can work well on some benchmarks, they can be complex and expensive: the model typically predicts high-dimensional observations, and determining the value of an action may require unrolling the model for multiple steps into the future.

In this paper, we show how we can estimate the environment dynamics in an efficient way while avoiding the dependence of model-based rollouts on the episode horizon. The model learned by our method *(1)* does not require predicting high-dimensional observations at every timestep, *(2)* directly predicts the distribution of future states without the need of autoregressive unrolls and *(3)* can be used

to estimate the value function without requiring expensive rollouts or temporal difference learning, nor does it need action or reward labels during the pre-training phase.

Precisely, we learn a generative model of the discounted state occupancy measure, i.e. a function which takes in a state, timestep and policy representation vector and returns a future state proportional to the likelihood of visiting that future state under some fixed policy. This occupancy measure resembles successor features (Dayan, 1993), and can be seen as its generative, normalized version. By scoring these future states by the corresponding rewards, we form an unbiased estimate of the value function. We name our proposed algorithm Diffused Value Function (DVF). Because DVF represents multi-step transitions implicitly, it avoids having to predict high-dimensional observations at every timestep and thus scales to long-horizon tasks with high-dimensional observations. Using the same algorithm, we can handle settings where reward-free and action-free data is provided, which cannot be directly handled by classical TD-based methods. Specifically, the generative model can be pre-trained on sequences of states without the need for reward or action labels, provided that some representation of the data generating process (i.e., behavior policy) is known.

We highlight the strengths of DVF both qualitatively and quantitatively on challenging robotic tasks, and show how generative models can be used to accelerate *tabula rasa* learning. Specifically, we conduct online experiments on continuous Mountain Car to demonstrate how DVF can adapt to non-stationary problems. Our results also suggest that DVF is a competitive method for offline continuous control tasks such as d4rl PyBullet (Erwin & Yunfei, 2016) and Franka Kitchen. Finally, we show how DVF can be used for learning exploration policies from offline datasets by perturbing the value function with a measure of long-term curiosity.

## 2 PRELIMINARIES

**Reinforcement learning** Let $M$ be a Markov decision process (MDP) defined by the tuple $M = \langle \mathcal{S}, S_0, \mathcal{A}, \mathcal{T}, r, \gamma \rangle$, where $\mathcal{S}$ is a state space, $S_0 \subseteq \mathcal{S}$ is the set of starting states, $\mathcal{A}$ is an action space, $\mathcal{T} = p(\cdot|s_t, a_t) : \mathcal{S} \times \mathcal{A} \to \Delta(\mathcal{S})$ is a one-step transition function[1], $r : \mathcal{S} \times \mathcal{A} \to [r_{\min}, r_{\max}]$ is a reward function and $\gamma \in [0,1)$ is a discount factor. The system starts in one of the initial states $s_0 \in S_0$. At every timestep $t > 0$, the policy $\pi : \mathcal{S} \to \Delta(\mathcal{A})$ samples an action $a_t \sim \pi(\cdot|s_t)$. The environment transitions into a next state $s_{t+1} \sim \mathcal{T}(\cdot|s_t, a_t)$ and emits a reward $r_t = r(s_t, a_t)$. The aim is to learn a Markovian policy $\pi(a \mid s)$ that maximizes the return, defined as discounted sum of rewards, over an episode of length $H$:

$$\max_{\pi \in \Pi} \mathbb{E}_{p_{0:H}^\pi, S_0} \left[ \sum_{t=0}^{H} \gamma^t r(s_t, a_t) \right], \tag{1}$$

where $p_{t:t+K}^\pi$ denotes the joint distribution of $\{s_{t+k}, a_{t+k}\}_{k=1}^K$ obtained by rolling-out $\pi$ in the environment for $K$ timesteps starting at timestep $t$. To solve Eq. (1), value-based RL algorithms estimate the future expected discounted sum of rewards, known as the *value function*:

$$Q^\pi(s_t, a_t) = \mathbb{E}_{p_t^\pi} \left[ \sum_{k=1}^{H} \gamma^{k-1} r(s_{t+k}, a_{t+k}) | s_t, a_t \right], \tag{2}$$

for $s_t \in \mathcal{S}, a_t \in \mathcal{A}$, and $V^\pi(s_t) = \mathbb{E}_\pi[Q(s_t, a_t)]$. Alternatively, the value function can be written as the expectation of the reward over the discounted occupancy measure:

$$Q^\pi(s_t, a_t) = \frac{1 - \gamma^{H-t}}{1 - \gamma} \mathbb{E}_{s, a \sim \rho^\pi(s_t, a_t), \pi(s)} [r(s, a)] \tag{3}$$

where $\rho^\pi(s|s_t, a_t) = (1 - \gamma) \sum_{\Delta t=1}^{H} \gamma^{\Delta t - 1} \rho^\pi(s|s_t, a_t, \Delta t, \pi)$ and $\rho^\pi(s|s_t, a_t, \Delta t, \pi) = \mathbb{P}[S_{t+\Delta t} = s|s_t, a_t; \pi]$ as defined in Janner et al. (2020)[2]. Most importantly, implicit conditioning of $\rho$ on $\pi$ is a hard task in itself (Harb et al., 2020; Mazoure et al., 2022a). We introduce a version of $\rho$ explicitly conditioned on some latent representation of the policy $\phi(\pi) : \Pi \to \mathbb{R}^d$. We denote this as $\rho(s|s_t, a_t, \Delta t, \phi(\pi)) = \rho^\pi(s|s_t, a_t, \Delta t)$. We discuss how to pick a reasonable $\phi$ in Section 3.1.

This decomposition of the value function has been shown to be useful in previous works based on the successor representation (Dayan, 1993; Barreto et al., 2016) and $\gamma$-models (Janner et al., 2020), and we will leverage this formulation to build a diffusion-based estimate of the value function below.

---

[1]$\Delta(\mathcal{X})$ denotes the entire set of distributions over the space $\mathcal{X}$.

[2]The random offset $\Delta t$ determines the model's predictive horizon in a multi-step setting.

**Diffusion models** Diffusion models form a class of latent variable models (Sohl-Dickstein et al., 2015) which represent the distribution of the data as an iterative process:

$$\mathbf{x}_0 \sim p(\mathbf{x}_0) = \mathbb{E}_{p_\theta(\mathbf{x}_{1:T})}[p_\theta(\mathbf{x}_0|\mathbf{x}_{1:T})] = p(\mathbf{x}_T)\prod_{t_d=1}^{T} p_\theta(\mathbf{x}_{t_d-1}|\mathbf{x}_{t_d}), \tag{4}$$

for $T$ latents $\mathbf{x}_{1:T}$ with conditional distributions parameterized by $\theta$. Here $t_d$ represents the time-step of the iterative process (not to be confused with the episode timestep $t$). The joint distribution of data and latents factorizes into a Markov Chain with parameters

$$p_\theta(\mathbf{x}_{t_d-1}|x_{t_d}) = \mathcal{N}(\mu_\theta(\mathbf{x}_{t_d},t_d),\Sigma_\theta(\mathbf{x}_{t_d},t_d))), \quad \mathbf{x}_T \sim \mathcal{N}(\mathbf{0},\mathbf{I}) \tag{5}$$

which is called the *reverse* process. The posterior $q(\mathbf{x}_{1:T}|\mathbf{x}_0)$, called the *forward* process, typically takes the form of a Markov Chain with progressively increasing Gaussian noise parameterized by variance schedule $\beta(t_d)$:

$$q(\mathbf{x}_{1:T}|\mathbf{x}_0) = \prod_{t_d=1}^{T} q(\mathbf{x}_{t_d}|\mathbf{x}_{t_d-1}), \quad q(\mathbf{x}_{t_d}|\mathbf{x}_{t_d-1}) = \mathcal{N}(\sqrt{1-\beta(t_d)}\mathbf{x}_{t_d-1},\beta(t_d)\mathbf{I})) \tag{6}$$

where $\beta$ can be either learned or fixed as hyperparameter. The parameters $\theta$ of the reverse process are found by minimizing the variational upper-bound on the negative log-likelihood of the data:

$$\mathbb{E}_q\left[-\log p(\mathbf{x}_T) - \sum_{t_d=1}^{T}\log\frac{p_\theta(\mathbf{x}_{t_d-1}|\mathbf{x}_{t_d})}{q(\mathbf{x}_{t_d}|\mathbf{x}_{t_d-1})}\right] \tag{7}$$

Later works, such as Denoising Diffusion Probabilistic Models (DDPM, Ho et al., 2020) make specific assumptions regarding the form of $p_\theta$, leading to the following simplified loss with modified variance scale $\bar{\alpha}(t_d) = \prod_{s=1}^{t_d}(1-\beta(s))$:

$$\ell_{\text{Diffusion}}(\theta) = \mathbb{E}_{\mathbf{x}_0,t_d,\epsilon}\left[||\epsilon - \epsilon_\theta(\sqrt{\bar{\alpha}(t_d)}\mathbf{x}_0 + \sqrt{1-\bar{\alpha}(t_d)}\epsilon,t_d)||_2^2\right],$$

$$\mathbf{x}_0 \sim q(\mathbf{x}_0), \quad t_d \sim \text{Uniform}(1,T), \quad \epsilon \sim \mathcal{N}(\mathbf{0},\mathbf{I}) \tag{8}$$

by training a denoising network $\epsilon_\theta$ to predict noise $\epsilon$ from a corrupted version of $\mathbf{x}_0$ at timestep $t_d$. Samples from $p(\mathbf{x}_0)$ can be generated by following the reverse process:

$$\mathbf{x}_{t_d-1} = \frac{1}{\sqrt{\alpha(t_d)}}\left(\mathbf{x}_{t_d} - \frac{1-\alpha(t_d)}{\sqrt{1-\bar{\alpha}}}\epsilon_\theta(\mathbf{x}_{t_d},t_d)\right) + \sigma_{t_d}\mathbf{z}, \quad \mathbf{x}_T \sim \mathcal{N}(\mathbf{0},\mathbf{I}),\mathbf{z} \sim \mathcal{N}(\mathbf{0},\mathbf{I}). \tag{9}$$

## 3 METHODOLOGY

Through the lens of Eq. (3), the value function can be decomposed into three components: (1) occupancy measure $\rho^\pi(s)$, dependent on **states** and **policy** and parameterized by a diffusion model, (2) reward model $r(s,a)$ dependent on **states** and **actions** and (3) policy representation $\phi(\pi)$, dependent on the **policy**. Equipped with these components, we could estimate the value of any given policy in a zero-shot manner. However, two major issues arise:

- For offline[3] training, $\rho^\pi$ has to be *explicitly* conditioned on the target policy, via the policy representation $\phi(\pi)$. That is, rather than estimating the occupation measure through online Monte Carlo rollouts, we choose to learn a direct model which maps some policy representation to its occupancy measure.
- Maximizing $Q(s,a,\phi(\pi))$ directly as opposed to indirectly via $r(s,a) + \gamma\mathbb{E}[V(s',\phi(\pi))]$ is too costly due to the large size of diffusion denoising networks. Specifically, if $\epsilon_\theta$ is a cross-attentional architecture like Perceiver, computing $\nabla_a Q(s,a,\phi(\pi))$ involves computing $\nabla_a \epsilon_\theta(s,a,\phi(\pi))$.

If both challenges are mitigated, then the value function $V^\pi(s_t)$ can be estimated by first sampling a collection of $n$ states from the learned diffusion model $s_{t+\Delta t,1},..,s_{t+\Delta t,n} \sim \rho^\pi(s_t)$ and then evaluating the reward predictor at those states $\sum_{i=1}^{n} r(s_{t+\Delta t,i},\pi(s_{t+\Delta t,i})) \propto V^\pi(s_t)$. A similar result can be derived for the state-action value function by training a state-action conditioned diffusion model $\rho^\pi(s_t,a_t)$, from which a policy can be decoded using e.g. the information projection method such as in Haarnoja et al. (2018).

---

[3]Online training can use implicit conditioning by re-collecting data with the current policy $\pi$

Figure 1: **Three crucial components of DVF.** *(left)* construct tuples $(s_t, s_{t+1}, s_{t+\Delta t})$ for training the diffusion model; *(middle)* architecture of the diffusion model, which takes in future noisy state $x$, current state $s_t$, time offset $\Delta t$, policy embedding $\phi(\pi)$ and diffusion timestep $t_d$ and processes them using the Perceiver I/O architecture (Jaegle et al., 2021) to predict the noise; *(right)* Sampling mechanism based on DPPM (Ho et al., 2020) is used with a reward model to estimate the value function

### 3.1 CHALLENGE 1: OFF-POLICY EVALUATION THROUGH CONDITIONING

Explicit policy conditioning has been (and still remains) a hard task for reinforcement learning settings. Assuming that the policy $\pi$ has a lossless finite-dimensional representation $\phi(\pi)$, passing it to an ideal value function network as $Q(s, a, \phi(\pi))$ could allow for zero-shot policy evaluation. That is, given two policy sets $\Pi_1, \Pi_2 \subseteq \Pi$, training $Q(s, a, \phi(\pi))$ on $\{s, a, \phi(\pi)\}, \pi \in \Pi_1$ and then swapping out $\phi(\pi)$ for $\phi(\pi')$ where $\pi' \in \Pi_2$ would immediately give the estimate of $Q^{\pi'}$.

We address this issue by studying sufficient statistics of $\pi$. Since the policy is a conditional distribution, it is possible to use a kernel embedding for conditional distributions such as a Reproducing Kernel Hilbert Space (Song et al., 2013; Mazoure et al., 2022a), albeit it is ill-suited for high-dimensional non-stationary problems. Recent works have studied using the trajectories $\{s_i, a_i\}_i^n$ as a sufficient statistic for $\pi$ evaluated at *key* states $s_1, .., s_n$ (Harb et al., 2020). Similarly, we studied two policy representations:

1. **Scalar:** Given a countable policy set $\Pi$ indexed by $i = 1, 2, ..$, we let $\phi(\pi) = i$. One example of such sets is the value improvement path, i.e. the number of training gradient steps performed since initialization.
2. **Sequential:** Inspired by Song et al. (2013), we embed $\pi$ using its rollouts in the environment $\{s_i, a_i\}_i^n$. In the case where actions are unknown, then the sequence of states can be sufficient, under some mild assumptions[4], for recovering $\pi$.

Both representations have their own advantages: scalar representations are compact and introduce an ordering into the policy set $\Pi$, while sequential representations can handle cases where no natural ordering is present in $\Pi$ (e.g. learning from offline data). Since we implement DVF on top of the Perceiver I/O architecture (Jaegle et al., 2021), $\phi(\pi)$ can be thought of as the context fed to the encoder.

### 3.2 CHALLENGE 2. MAXIMIZING THE VALUE WITH LARGE MODELS

In domains with continuous actions, the policy is usually decoded using the information projection onto the value function estimate (see Haarnoja et al. (2018)) by minimizing

$$\ell_{\text{Policy}}(\phi^5) = \mathbb{E}_{s \sim \mathcal{D}} \left[ \text{KL}\left( \pi_\phi(\cdot | s) || \frac{e^{Q^{\pi_{\text{old}}}(s, \cdot)}}{\sum_{a'} e^{Q^{\pi_{\text{old}}}(s, a')}} \right) \right]. \tag{10}$$

However, (a) estimating $Q^*(s, a)$ requires estimation of $\rho^*(s, a)$ which cannot be pre-trained on videos (i.e. state sequences) and (b) requires the differentiation of the sampling operator from the $\rho$ network, which, in our work, is parameterized by a large generative model, Perceiver IO (Jaegle et al., 2021). The same problem arises in both model-free (Haarnoja et al., 2018) and model-based methods (Hafner et al., 2023), where the networks are sufficiently small that the overhead is minimal. In our work, we circumvent the computational overhead by unrolling one step of Bellman backup

$$Q^\pi(s_t, a_t) = r(s_t, a_t) + \gamma \mathbb{E}_{s_{t+1}}[V^\pi(s_{t+1})] \tag{11}$$

and consequently

$$\nabla_{a_t} Q^\pi(s_t, a_t) = \nabla_{a_t} r(s_t, a_t) + \frac{1 - \gamma^{H-t-1}}{1 - \gamma} \sum_{i=1}^{n} \nabla_{a_t} r(s_{t+1+\Delta t, i}, a_{t+1+\Delta t, i}), \tag{12}$$

---

[4]One such case is MDPs with deterministic dynamics, as it allows to figure out the corresponding action sequence.

[5]The policy representation $\phi(\pi)$ overloads the notation for policy parameters $\phi$.

---

**Algorithm 1:** Diffused Value Function (DVF)

---

**Input** :Behavior
policy $\mu$ and corresponding dataset $\mathcal{D} \sim \mu$, $\epsilon_\theta, r_\psi, \pi_\phi$ networks, number of samples $n$

```
/* Normalize states from D to lie in [−1,1] interval      */
```
1 $\mathcal{D}[s] \leftarrow \frac{\mathcal{D}[s] - \min \mathcal{D}[s]}{\max \mathcal{D}[s] - \min \mathcal{D}[s]}$ ;

2 **for** *epoch* $j = 1, 2, .., J$ **do**

3 $\quad$ **for** *minibatch* $\mathcal{B} \sim \mathcal{D}$ **do**

```
      /* Construct policy context                          */
```
4 $\quad\quad$ Construct $\phi(\pi)$ using either the scalar or sequential approach ;

```
      /* Update diffusion model ρθusing Eq. (8)           */
```
5 $\quad\quad$ Update $\epsilon_\theta$ using $\nabla_\theta \ell_{\text{Diffusion}}(\theta^{(j)})$ using $s_t, s_{t+\Delta t}, \phi(\pi) \sim \mathcal{D}$;

```
      /* Update the reward estimator                       */
```
6 $\quad\quad$ Update $r_\psi$ using $\nabla_\psi \mathbb{E}_{s,a} \left[ ||r_\psi(s,a) - r(s,a)||_2^2 \right]$ ;

```
      /* Estimate V                                        */
```
7 $\quad\quad$ $V(s_{t+1}) \leftarrow \frac{1-\gamma^{H-t-1}}{1-\gamma} \sum_{i=1}^n r(s_{t+1+\Delta t,i}, \pi_\phi(s_{t+1+\Delta t,i})), s_{t+1+\Delta t} \overset{\text{DDPM}}{\sim} \rho_\theta(s_{t+1}, \phi(\pi))$;

```
      /* Estimate Q                                        */
```
8 $\quad\quad$ $Q(s_t, a_t) \leftarrow r(s_t, a_t) + \gamma V(s_{t+1})$ ;

```
      /* Decode policy from Q-function using Eq. (10)      */
```
9 $\quad\quad$ Update $\pi_\phi$ using $\nabla_\phi \ell_{\text{Policy}}(\phi)$ and $Q(s_t, a_t)$ ;

---

allowing to learn $\rho^\pi(s)$ instead of $\rho^\pi(s,a)$ and using it to construct the state value function. Moreover, differentiating the small reward network is more computationally efficient than computing the gradient over the fully-attentional diffusion model.

### 3.3 PRACTICAL ALGORITHM

As an alternative to classical TD learning, we propose to separately estimate the occupancy measure $\rho$, using a denoising diffusion model and the reward $r$, using a simple regression in symlog space (Hafner et al., 2023). While it is hard to estimate the occupancy measure $\rho$ directly, we instead learn a denoising diffusion probabilistic model $\epsilon_\theta$ (DDPM, Ho et al., 2020), which we call the *de-noising network*. Since we know what the true forward process looks like at diffusion timestep $t_d$, the de-noising network $\epsilon_\theta : \mathcal{S} \to [-1,1]$ is trained to predict the input noise.

The high-level idea behind the algorithm is as follows:

1. Pre-train a diffusion model on sequences of states $s_1, .., s_H$ and, optionally, policy embeddings $\phi(\pi)$. We use classifier-free guidance to condition the diffusion model, by passing $x = [\sqrt{\bar{\alpha}(t_d)} \mathbf{x}_0 + \sqrt{1 - \bar{\alpha}(t_d)} \epsilon, s_t, t_d, \phi(\pi)]$ to the de-noising network $\epsilon_\theta$. This step can be performed on large amounts of demonstration videos without the need of any action nor reward labels. The policy embedding $\phi(\pi)$ can be chosen to be any auxiliary information which allows the model to distinguish between policies[6]. This step yields $\rho(s_t; \phi(\pi))$.
2. Using labeled samples, train a reward predictor $r(s,a)$. This reward predictor will be used as importance weight to score each state-action pair generated by the diffusion model.
3. Sample a state from $\rho(\cdot, \phi(\pi))$ and score it using $r(s_t, \pi(s_t))$, thus obtaining an estimate proportional to the value function of policy $\pi$ at state $s_t$.
4. *Optionally:* Maximize the resulting value function estimator using the information projection of $\pi$ onto the polytope of value functions (see Eq. (10)) and decoding a new policy $\pi'$. If in the online setting, use $\pi'$ to collect new data in the environment and update $\phi(\pi)$ to $\phi(\pi')$. In the offline setting, use autoregressive rollouts in $\rho$ to form the new context $\phi(\pi')$.

Algorithm 1 describes the exact training mechanism, which first learns a diffusion model and maximizes its reward-weighted expectation to learn a policy suitable for control. Note that the method is suitable for both online and offline reinforcement learning tasks, albeit conditioning on the policy representation $\phi(\pi)$ has to be done explicitly in the case of offline RL. DVF can also be shown to learn

---

[6]In MDPs with deterministic transitions, one such example is $s_1, .., s_t$.

an occupancy measure which corresponds to the normalized successor features (Dayan, 1993) that allows sampling future states through the reverse diffusion process (see Janner et al. (2020) for proof).

### 3.4 ANALYSIS

A major problem with Monte-Carlo offline RL approaches is that, without explicit policy conditioning, optimizing Eq. (8) over a fixed dataset $\mathcal{D}^\mu$ generated by logging policy $\mu$ yields $\rho^\mu$ and consequently $Q^\mu$. As per Lemma 6.1 of Kakade & Langford (2002), only a single step of policy improvement is possible without collecting new data. However, policy conditioning allows to estimate $Q^\pi$ off-policy, without access to an online simulator. Picking greedily

$$\pi_{i+1} = \operatorname*{argmax}_{\pi \in \Pi} \mathbb{E}_{(s,a) \sim \rho^{\pi_i}, \pi_i}[Q^\pi(s, \pi(s))] \text{ where } \pi_0 = \mu \tag{13}$$

guarantees policy improvement since, as per the performance difference lemma,

$$J(\pi_{i+1}) - J(\pi_i) = \frac{1}{\gamma - 1} \mathbb{E}_{(s,a) \sim \rho^{\pi_i}, \pi_i}[A^{\pi_{i+1}}(s, a)] \geq 0 \tag{14}$$

for $J(\pi) = \mathbb{E}_{s_0}[V^\pi(s_0)]$, $A^\pi(s, a) = Q^\pi(s, a) - V^\pi(s)$ and policy index $i \geq 0$. Note that performance improvement can be achieved by either maximizing $\mathbb{E}_{(s,a) \sim \rho^{\pi_i}, \pi_i}[A^{\pi_{i+1}}(s, a)]$ or $\mathbb{E}_{(s,a) \sim \rho^{\pi_{i+1}}, \pi_{i+1}}[A^{\pi_i}(s, a)]$. The first formulation has to be used due to constraints of offline RL and the ability of DVF to estimate off-policy value functions. In practice, after the diffusion model is trained with $\phi(\mu)$, we pass $\phi(\pi_i)$ to perform policy conditioning. For the case of sequential policy embeddings, we perform autoregressive rollouts of $\pi_i$ in the diffusion model, similar to DynaQ (Sutton, 1991).

## 4 EXPERIMENTS

### 4.1 MOUNTAIN CAR

Before studying the behavior of DVF on robotic tasks, we conduct experiments on the continuous Mountain Car problem, a simple domain for analysing sequential decision making methods. We trained DVF for 500 gradient steps until convergence, and computed correlations between the true environment returns, the value function estimator based on the diffusion model, as well as the reward prediction at states sampled from $\rho^\pi$. Fig. 2 shows that all three quantities exhibit positive correlation, even though the value function estimator is not learned using temporal difference learning.

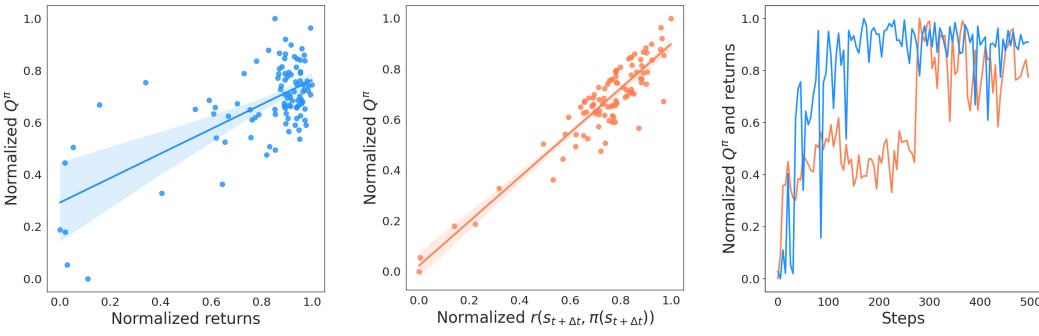

Figure 2: **Value-reward-returns correlation.** *(Left)* Returns positively correlate with the diffused value function, *(Middle)* Diffused value function strongly correlates with future reward and (Right) diffused value function (orange) and returns (blue) increase as training progresses.

### 4.2 MAZE 2D

We examine the qualitative behavior of the diffusion model of DVF on a simple locomotion task inside mazes of various shapes, as introduced in the D4RL offline suite (Fu et al., 2020). In these experiments, the agent starts in the lower left of the maze and uses a waypoint planner with three separate goals to collect data in the environment (see Fig. 3(a) and Fig. 3(c) for the samples of the collected data). The diffusion model of DVF is trained on the data from the three data-collecting policies, using the scalar policy conditioning described in Section 3.1.

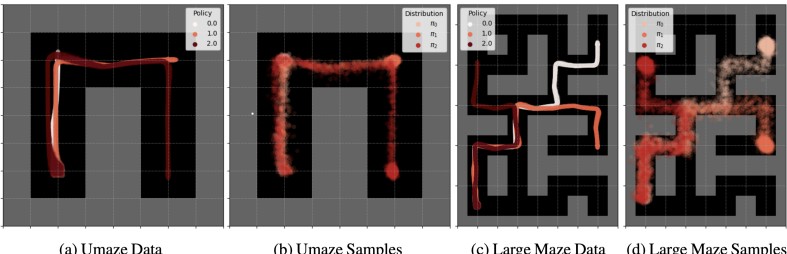

|     |     |     |     |
| --- | --- | --- | --- |
| (a) Umaze Data | (b) Umaze Samples | (c) Large Maze Data | (d) Large Maze Samples |

Figure 3: **Effect of policy conditioning.** *(a, c)* Ground truth data distribution for the u-maze and large maze from the Maze 2d environment. *(b, d)* Conditional distribution of future states $\rho(s_{t+\Delta t}|s_0, \phi(\pi_i))$ given the starting state in the bottom left corner and the policy index. The diffusion model correctly identifies and separates the three state distributions in both mazes.

Fig. 3 shows full trajectories sampled by conditioning the diffusion model on the start state in the lower left, the policy index, and a time offset. Fig. 4 shows sampled trajectories as the discount factor $\gamma$ increases, leading to sampling larger time offsets.

The results show the ability of the diffusion model to represent long-horizon data faithfully, and highlight some benefits of the approach. DVF can sample trajectories without the need to evaluate a policy or specify intermediate actions. Because DVF samples each time offset independently, there is also no concern of compounding model error as the horizon increases. Additionally, the cost of predicting $s_{t+k}$ from $s_t$ is $\mathcal{O}(1)$ for DVF, while it is $\mathcal{O}(k)$ for classical autoregressive models.

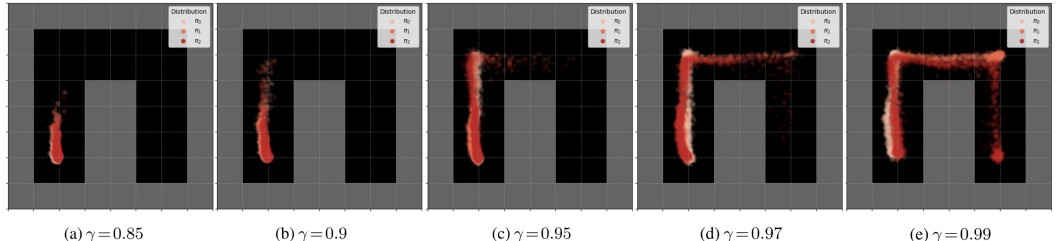

|     |     |     |     |     |
| --- | --- | --- | --- | --- |
| (a) $\gamma = 0.85$ | (b) $\gamma = 0.9$ | (c) $\gamma = 0.95$ | (d) $\gamma = 0.97$ | (e) $\gamma = 0.99$ |

Figure 4: **Effect of discount factor.** Samples from the learned diffusion model with increasing discount factor $\gamma$, with a starting state in the lower left of the maze. As $\gamma$ increases, the model generates samples further along the trajectory leading to the furthest point of the maze. Ground truth data shown in Fig. 3(a)

Note that, for the case of DVF, using an autoregressive strategy to predict the sequential policy embedding $\phi(\pi)$ scales well, as its cost depends on the context length $n$ instead of future prediction horizon $H$, and in deterministic MDPs it is often the case that $n << H$.

### 4.3 CONTROL FROM OFFLINE DEMONSTRATIONS

**PyBullet** In order to test our algorithm in more realistic settings, our next set of experiments consists in ablations performed on offline data collected from classical PyBullet environments (Körber et al., 2021)[7]. We compare DVF to behavior cloning and Conservative Q-learning (Kumar et al., 2020), two strong offline RL baselines. We also plot the average normalized returns of each dataset to facilitate the comparison over 5 random seeds. Medium dataset contains data collected by medium-level policy and mixed contains data from SAC (Haarnoja et al., 2018) training.

Fig. 5 highlights the ability of DVF to match and sometimes outperform the performance of classical offline RL algorithms, especially on data from a random policy. In domains where online rollouts can be prohibitively expensive, the ability to learn from incomplete offline demonstrations is a strength of DVF. This benchmark also demonstrates the shortcomings of scalar policy representation, which is unknown for a given offline dataset, and also doesn't scale well when the number of policies is large (e.g. $\phi(\pi_{(j)}) = j$ after taking $j$ gradient steps to obtain $\pi_{(j)}$ from $\pi_0$).

---

[7]Data is taken from https://github.com/takuseno/d4rl-pybullet/tree/master

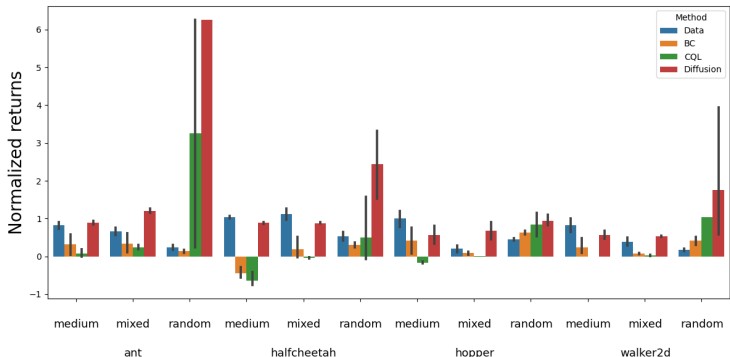

Figure 5: **PyBullet performance.** Returns obtained by our diffusion model, behavior cloning, CQL on 4 challenging robotic tasks from the PyBullet offline suite, together with average returns in each dataset (*Data* in the plot). Returns are normalized by average per-task data performance.

| Task | BC | CQL | IQL | BRAC-p | BEAR | Diffusion BC | Diffusion QL | DVF | DVF (pooled) |
|------|-----|-----|-----|--------|------|--------------|--------------|-----|--------------|
| kitchen-complete-v0 | 65.0 | 43.8 | 62.5 | 0.0 | 0.0 | $53.8 \pm 10$ | $67.2 \pm 8$ | $74.8 \pm 13$ | $79.1 \pm 12$ |
| kitchen-partial-v0 | 38.0 | 49.8 | 46.3 | 0.0 | 0.0 | $42.5 \pm 3$ | $48.1 \pm 4$ | $50.2 \pm 7$ | $72.2 \pm 9$ |
| kitchen-mixed-v0 | 51.5 | 51.0 | 51.0 | 0.0 | 0.0 | $48.3 \pm 2$ | $53.5 \pm 6$ | $51.1 \pm 7$ | $67.4 \pm 6$ |

Table 1: **Franka Kitchen.** Returns collected by RL agents on D4RL's Franka Kitchen tasks, computed on 5 random seeds. Diffusion BC refers to Chi et al. (2023), while Diffusion QL refers to Wang et al. (2022).

| Task | BC | CQL | SAC | TD3+BC | DVF |
|------|-----|-----|-----|--------|-----|
| maze2d-umaze | $1.07 \pm 0.35$ | $1.00 \pm 0.43$ | $1.00 \pm 0.23$ | $1.00 \pm 0.23$ | $125.04 \pm 32.57$ |
| maze2d-medium | $1.00 \pm 0.27$ | $5.00 \pm 4.10$ | $5.00 \pm 3.61$ | $1.00 \pm 0.31$ | $30.21 \pm 13.42$ |
| maze2d-large | $5.00 \pm 3.8$ | $1.08 \pm 0.32$ | $1.00 \pm 0.35$ | $3.00 \pm 1.36$ | $4.14 \pm 4.02$ |

Table 2: **Exploration from offline data.** Number of times the offline agent solves each sparse reward task over 10,000 training steps. Mean $\pm$ standard deviation are computed on 5 random seeds.

**Franka Kitchen** We compare the performance of DVF to that of existing algorithms on the challenging Franka Kitchen domain from D4RL. Baseline numbers are taken directly from Kostrikov et al. (2021). We also include DVF (pooled) for comparison, which pre-trains the diffusion denoising network $\epsilon_\theta$ on all datasets in the task suite, and only uses the labeled transitions from the evaluation task to learn the reward and policy networks. Table 1 shows that, on this complex set of tasks, DVF performs competitively with existing offline RL methods, and that data pooling can help the diffusion model learn the policy conditioning operator. The Diffusion BC and Diffusion QL baselines are allowed $\times 3$ computational budget (i.e. 300k gradient steps over 100k).

## 4.4 EXPLORATION FROM OFFLINE DATA

An important problem rarely addressed by offline RL methods is learning good exploration policies. If the offline demonstrations are different enough from the online task, the value function learned by offline RL methods is of little use directly, due to behavior regularization. Instead, learning a value function which balances extrinsic rewards with a curiosity objective gives rise to policies which cover a larger fraction of the state space. We modify all algorithms to instead maximize the compound reward

$$r_{\text{Exploration}}(s_t, a_t) = r(s_t, a_t) + \alpha \mathbb{E}_{s_{t+\Delta t} \sim \rho^\pi(s_t)}[||f(s_t) - f(s_{t+\Delta t})||], \tag{15}$$

where $\alpha > 0$ controls the exploration rate and $f$ projects states onto a latent space. The coefficient $\alpha$ can be annealed to balance exploration with maximizing extrinsic rewards found in the offline dataset. For all methods except DVF, $\Delta t = 1$, i.e. we used a curiosity bonus similar to RIDE (Raileanu & Rocktäschel, 2020), but using the algorithm's encoder as $f$. For DVF, we used the diffusion model's predictions.

Table 2 shows that using the future state occupancy model helps in sparse reward tasks. Figure 6 shows that using the diffusion model learned by DVF maximizes the state coverage in the 2d maze environment. Even in the case when no rewards are available (e.g. demonstrations contain only partial expert trajectories), the diffusion model can be used to simulate long-horizon rollouts, unlike single-step models.

## 5 Related works

**Offline pre-training for reinforcement learning**    Multiple approaches have tried to alleviate the heavy cost of training agents *tabula rasa* (Rigter et al., 2022; Swazinna et al., 2022; Fujimoto & Gu, 2021; Emmons et al., 2021) by pre-training the agent offline. For example, inverse dynamics models which predict the action between the current and next state have seen success in complex domains such as Atari (Schwarzer et al., 2021), maze environments (Ajay et al., 2020) as well as Minecraft (Baker et al., 2022; Fan et al., 2022). Return-conditioned sequence models have also seen a rise in popularity due to their ability to learn performance-action-state correlations over long horizons (Lee et al., 2022), but they learn point estimates and require reward and action information during training.

**Unsupervised reinforcement learning**    Using temporal difference (Sutton & Barto, 2018) for policy iteration or evaluation requires all data tuples to contain state, action and reward information. However, in some real-world scenarios, the reward might only be available for a small subset of data (e.g. problems with delayed feedback (Howson et al., 2021)). In this case, it is possible to decompose the value function into a reward-dependent and dynamics components, as was first suggested in the successor representation framework (Dayan, 1993; Barreto et al., 2016). More recent approaches (Janner et al., 2020; Eysenbach et al., 2020; 2022; Mazoure et al., 2022b) use a density model to learn the occupancy measure over future states for each state-action pair in the dataset. However, learning an explicit multi-step model such as (Janner et al., 2020) can be unstable due to the bootstrapping term in the temporal difference loss, and these approaches still require large amounts of reward and action labels. While our proposed method is a hybrid between model-free and model-based learning, it avoids the computational overhead incurred by classical world models such as Dreamer (Hafner et al., 2023) by introducing constant-time rollouts. The main issue with infinite-horizon models is the implicit dependence of the model on the policy, which imposes an upper-bound on the magnitude of the policy improvement step achievable in the offline case. Our work solves this issue by adding an explicit policy conditioning mechanism, which allows to generate future states from unseen policy embeddings.

**Diffusion models**    Learning a conditional probability distribution over a highly complex space can be a challenging task, which is why it is often easier to instead approximate it using a density ratio specified by an inner product in a much lower-dimensional latent space. To learn an occupancy measure over future states without passing via the temporal difference route, one can use denoising diffusion models to approximate the corresponding future state density under a given policy. Diffusion has previously been used in the static unsupervised setting such as image generation (Ho et al., 2020) and text-to-image generation (Rombach et al., 2022b). Diffusion models have also been used to model trajectory data for planning in small-dimensional environments (Janner et al., 2022), as well as for model-free (Wang et al., 2022; Ajay et al., 2022; Hansen-Estruch et al., 2023) and imitation (Chi et al., 2023) learning. However, no work so far has managed to efficiently predict infinite-horizon rollouts.

## 6 Discussion

In this work, we introduced a simple model-free algorithm for learning reward-maximizing policies, which can be efficiently used to solve complex robotic tasks. Diffused Value Function (DVF) avoids the pitfalls of both temporal difference learning and autoregressive model-based methods by pre-training an infinite-horizon transition model from state sequences using a diffusion model. This model does not require any action nor reward information, and can then be used to construct the state-action value function, from which one can decode the optimal action. DVF fully leverages the power of diffusion models to generate states far ahead into the future without intermediate predictions. Our experiments demonstrate that DVF matches and sometimes outperforms strong offline RL baselines on realistic robotic tasks for control and exploration from offline data, and opens an entire new direction of research.

## 7 Limitations

The main limitation of our method is that it operates directly on observations instead of latent state embeddings, which requires tuning the noise schedule for each set of tasks, instead of using a unified noise schedule similarly to latent diffusion models (Rombach et al., 2022a). Another limitation is the need to explicitly condition the rollouts from the diffusion model on the policy, something that single-step models avoid. Finally, online learning introduces the challenge of capturing the non-stationarity of the environment using the generative model $\rho$, which, in itself, is a hard task.

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

APPENDIX

## 7.1 EXPERIMENTAL DETAILS

**Model architecture**    DVF uses a Perceiver I/O model (Jaegle et al., 2021) with $1 \times 1$ convolution encodings for states, sinusoidal encoding for diffusion timestep and a linear layer for action embedding. The Perceiver I/O model has positional encodings for all inputs, followed by 8 blocks with 4 cross-attention heads and 4 self-attention heads and latent size 256. The scalar policy representation was encoded using sinusoidal encoding, while the sequential representation was passed through the $1 \times 1$ convolution and linear embedding layers and masked-out to handle varying context lengths, before being passed to the Perceiver model.

The policy and reward networks are represented as 4-layer MLPs with Mish activation function (Misra, 2019), Layernorm and residual connections.

| Hyperparameter | Value |
|---|---|
| Learning rate | $3 \times 10^{-4}$ |
| Batch size | 128 |
| Discount factor | 0.99 |
| Max gradient norm | 100 |
| MLP structure | $256 \times 256$ DenseNet MLP |
| Add LayerNorm in between all layers | Yes |

Table 3: Hyperparameters that are consistent between methods.

| Hyperparameter | Value |
|---|---|
| DVF | |
| Number of future state samples $n$ | 32 |
| BC coefficient | 1 |
| Context length $n$ | 4 |
| CQL | |
| Regularization coefficient | 1 |

Table 4: Hyperparameters that are different between methods.

All experiments were run on the equivalent of 2 V100 GPUs with 32 Gb of VRAM and 8 CPUs.

**Dataset composition**    The Maze2d datasets were constructed based on waypoint planning scripts provided in the D4RL repository, and modifying the target goal locations to lie in each corner of the maze (u-maze), or in randomly chosen pathways (large maze). The PyBullet dataset has a data composition similar to the original D4RL suite, albeit collected in the PyBullet simulator instead of MuJoCo.

## 7.2 ADDITIONAL RESULTS

We include three videos of the training of the diffusion model $\rho$ on the large maze dataset shown in Fig. 4 for 128, 512 and 1024 diffusion timesteps, in the supplementary material. Note that increasing the number of timesteps leads to faster convergence of the diffusion model samples to the true data distribution.

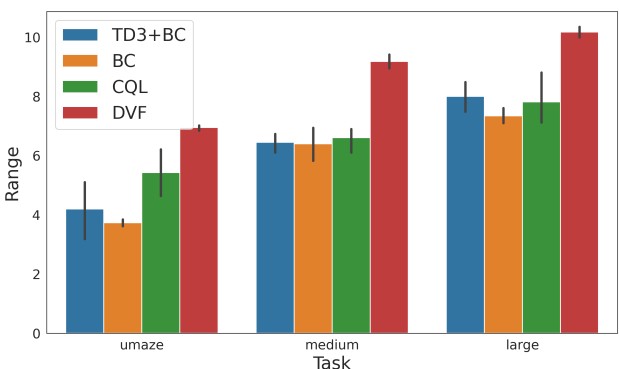

Figure 6: **Offline policy coverage.** Range (i.e. $\max(x) - \min(x)$) of states collected by DVF used as proxy measurement for state coverage. High range for DVF implies a higher exploration rate than classical offline RL methods.

