# OpenReview forum: "Value function estimation using conditional diffusion models for control"
_ICLR.cc/2024/Conference — Submitted to ICLR 2024_

### Official Review · Reviewer_ZNxJ · 2023-10-31

**Soundness:** 3 good
**Presentation:** 3 good
**Contribution:** 4 excellent
**Rating:** 8
**Confidence:** 4

**Summary:**

The paper proposes a paradigm for reinforcement learning that can be considered different from most prior approaches that either learn an auto-regressive transition model or a value function based on temporal difference learning. Instead, the authors develop a method that factories the components of a value function into three parts: The policy, a single step reward model, and a model of the state occupancies conditioned on the policy. Together, the components are able to estimate the value function by sampling states (and actions) from the occupancy model, scoring them with the reward model and training the policy to maximise these scores. The paper offers qualitative examination of the occupancy model on Maze2D data as well as empirical evaluation of the method on offline RL datasets from PyBullet, where it shows promising performances compared to the commonly known offline algorithm CQL. Further, it is shown that the occupancy model offers different perspectives for exploration based on offline data - by adding a reward term that encourages future states that are different from the current one, without the limitation of single-step models, the algorithm appears to perform very well in sparse reward settings.

**Strengths:**

To the best of my knowledge the paper presents a novel way of doing RL without temporal difference learning or dynamics modelling. Instead, by learning a diffusion based occupancy model of the states visited by the policy, some of the pitfalls of these methods seem to be effectively circumvented (e.g. accumulation of transition errors in model-based methods, usage of low quality data hard in temporal difference learning). The qualitative results look like the occupancy model is generally doing what is expected and the quantitative results show that the new paradigm is actually able to beat some well known algorithms on benchmark datasets for offline RL. Additionally, the occupancy model opens up new ways of performing exploration since it is not limited to a single time step.

**Weaknesses:**

In 3.1, the authors address the issue of conditioning the occupancy model on the policy. This appears to me like the hardest thing to do, especially for offline RL since we cannot test whether the model is correct without checking on the real environment. What we have is just a dataset where a behaviour policy (or maybe multiple) has collected interactions - as soon as we move away from replicating the policy(-ies) present in this dataset, we cannot really know the true occupancy and thus value estimation becomes tricky as well.

The authors propose 2 ways of conditioning on policies, either scalar (by enumerating the set of policies e.g. along the gradient steps) or sequential ("embed pi using its rollouts in the environment"). For both, it is a little ambiguous too me as to why they work:
In the offline RL comparison on the pybullet datasets, the latter option is chosen and I am wondering what that means, i.e. where do you get rollouts in the environment from the policy that you currently training and that's not the one that collected the dataset? It's not really offline RL then any more if you collect new rollouts, is it?
Similarly the scalar way: It is used in the qualitative experiments in maze2D, which makes perfect sense, but using it to embed policies along the improvement path appears like it could go very wrong as well since from one gradient update to the next the behaviour and thus the state-occupancy could change drastically.
Policy conditioning seems to me like one of the critical issues here, it seems you have made it work, so please share some more insights how and why.

One of the main empirical evaluations is done on the offline pybullet datasets. I understand the key contribution is showing that this novel method can work and not necessarily that it is currently the best one, however it seems to me the baselines are not particularly strong. I am surprised to see that CQL often even achieves negative returns, and outperforming BC and data performance especially on random/mixed datasets is also not particularly hard. I believe much value would be added to the manuscript if some more recent / successful offline RL algorithms were used as a comparison. Optimally you would include some model-based ones (e.g. [1,2]) as well as some model-free ones (e.g. [3,4]) since your method lies somewhere in between the two - adding a reward conditioned method that also relies neither on dynamics models nor TD, like RvS [5] could also be interesting. Also, since the main thing DVF is used for is offline RL, mentioning offline RL works like [1-5] in the related works section seems appropriate. Further, [6] could be an addition to the offline pre-training related work section.

[1] Rigter, M., Lacerda, B., & Hawes, N. Rambo-rl: Robust adversarial model-based offline reinforcement learning. NeurIPS 2022.

[2] Swazinna, P., Udluft, S., & Runkler, T. User-Interactive Offline Reinforcement Learning. ICLR 2023.

[3] Fujimoto, S., & Gu, S. S. A minimalist approach to offline reinforcement learning. NeurIPS 2021.

[4] Hansen-Estruch, P., Kostrikov, I., Janner, M., Kuba, J. G., & Levine, S. Idql: Implicit q-learning as an actor-critic method with diffusion policies. Preprint 2023.

[5] Emmons, S., Eysenbach, B., Kostrikov, I., & Levine, S. Rvs: What is essential for offline rl via supervised learning?. ICLR 2022.

[6] Ajay, A., Kumar, A., Agrawal, P., Levine, S., & Nachum, O. Opal: Offline primitive discovery for accelerating offline reinforcement learning. ICLR 2021.

**Questions:**

See weaknesses

---

> ### Author Response · Authors · 2023-11-17
> **Author response**
>
> We thank the reviewer for their thorough feedback on our paper, and their interesting questions. We provide our responses below:
>
> > Policy conditioning
>
> Since our work primarily tackles the challenges arising in offline settings, it is impossible to collect on-policy samples during policy optimization. We’ve experimented with two potential solutions to this issue. The first strategy is to simply condition the occupancy measure (i.e. noise network in diffusion) on data available in the training set. This approach maximizes the value of the behavior policy, and yields a single step of policy improvement which, in most realistic MDPs, is enough to find a policy with higher returns [1,2]. The second strategy is re-generate the policy embedding $\phi(\pi)$ for the learned policy, which can be done with continuous embeddings by using $\rho$ and $\pi$ to predict the future states in an autoregressive manner (see end of Section 3.4). Since the context for deterministic environments like MuJoCo is rather small, this doesn’t pose a significant overhead.
> Policy conditioning has been extensively studied by works treating RL problems as reward-conditioned sequence modelling [3,4], and we heavily borrowed architectural design choices from this literature, notably the use of transformers (specifically, Perceiver IO due to its ability to accept arbitrary input-output modalities) as opposed to MLPs, which primarily used in model-free RL.
>
> > Comparison to other diffusion-based baselines
>
> We agree that our experimental section could benefit from additional baselines that are also based on generative models such as [6,7]. We present the results below, as well as in Table 1 of the revised manuscript.
>
> | **Task**                | **BC** | **CQL** | **IQL** | **BRAC-p** | **BEAR** | **Diffusion BC** | **Diffusion QL** | **DVF**       | **DVF (pooled)** |
> |:-----------------------:|:------:|:-------:|:-------:|:----------:|:--------:|:----------------:|:----------------:|:-------------:|:----------------:|
> | **kitchen-complete-v0** | 65.0   | 43.8    | 62.5    | 0.0        | 0.0      | 53.8 $\pm$ 10    | 67.2 $\pm$ 8     | 74.8 $\pm$ 13 | 79.1 $\pm$ 12    |
> | **kitchen-partial-v0**  | 38.0   | 49.8    | 46.3    | 0.0        | 0.0      | 42.5 $\pm$ 3     | 48.1 $\pm$ 4     | 50.2 $\pm$ 7  | 72.2 $\pm$ 9     |
> | **kitchen-mixed-v0**    | 51.5   | 51.0    | 51.0    | 0.0        | 0.0      | 48.3 $\pm$ 2     | 53.5 $\pm$ 6     | 51.1 $\pm$ 7  | 67.4 $\pm$ 6     |
>
>
> The new baselines based on diffusion roughly match the performance of DVF, while being trained on x3 more compute. Note that we had to re-run the author’s code for [2] in our setting, specifically using a single set of hyper-parameters for all runs.
>
> > Missing references
>
> We thank the reviewer for their thoroughness, and added the suggested missing references to Section 5 of the revised paper.
>
> ## References:
>
> [1] Kakade, Sham, and John Langford. "Approximately optimal approximate reinforcement learning." Proceedings of the Nineteenth International Conference on Machine Learning. 2002.
>
> [2] Brandfonbrener, David, et al. "Offline rl without off-policy evaluation." Advances in neural information processing systems 34 (2021): 4933-4946.
>
> [3] Chen, Lili, et al. "Decision transformer: Reinforcement learning via sequence modeling." Advances in neural information processing systems 34 (2021): 15084-15097.
>
> [4] Janner, Michael, Qiyang Li, and Sergey Levine. "Offline reinforcement learning as one big sequence modeling problem." Advances in neural information processing systems34 (2021): 1273-1286.
>
> [5] Chi, Cheng, et al. "Diffusion policy: Visuomotor policy learning via action diffusion." arXiv preprint arXiv:2303.04137 (2023).
>
> [7] Wang, Zhendong, Jonathan J. Hunt, and Mingyuan Zhou. "Diffusion policies as an expressive policy class for offline reinforcement learning." arXiv preprint arXiv:2208.06193(2022).

---

### Official Review · Reviewer_nnB2 · 2023-10-31

**Soundness:** 2 fair
**Presentation:** 2 fair
**Contribution:** 1 poor
**Rating:** 3
**Confidence:** 4

**Summary:**

This paper proposes to train a diffusion model for estimating the state occupancy measure $\rho(s,a)$ as well as a reward model $r(s,a)$ and uses these networks to train a policy $\pi$ to solve a given task. The authors evaluate their method on a slate of offline RL tasks and show improvement over prior works in offline RL.

**Strengths:**

Strengths:
* proposes a novel application of Diffusion Models to offline RL - instead of training a model for dynamics prediction, train it for state occupancy prediction and use that to compute the reward function without learning a value function directly (DVF)
* proposes conditioning the diffusion model on the policy embedding which enables it generate future states from unseen policy embeddings
* method outperforms BC and CQL on d4rl benchmark tasks

**Weaknesses:**

I have serious concerns regarding the position and framing of this paper as well as the experiments. This work is written as if there is little, if any work in applying Diffusion Models in the offline RL/BC setting, citing only Diffuser (Janner et al) while failing to note Diffusion-QL (Wang et al), AdaptDiffuser (Liang et al.), Diffusion Policy (Chi et al) and many more works. The introduction, related works and methods section are all missing this crucially important context to properly understand the contribution. Writing-wise, the methods section is also extremely difficult to follow - there are many typos, notation mistakes and a math error (specifically equation 12 is wrong, there needs to be a term with the dynamics as well). Finally, the authors fail to compare against any Diffusion-based baselines in their work, which would lead the reader to believe that the proposed DVF method is a state-of-the-art method for doing offline RL. As a simple example, see Table 1 in the Diffusion-QL paper - DVF (non-pooled, which is the fair comparison) performs worse than Diffusion-QL in every task. It also appears that even with the curiosity reward added to the offline RL datasets, the Maze2d results (Table 2) are worse than those in Table 1 of the Diffuser paper.

Notes:
* Figure 1, $s_{t+\Delta}$ is used multiple times in the leftmost picture - they should have different subscripts to denote they are using different deltas
* The method description completely skips describing the averaging step that is necessary to get a state occupancy estimate that is not dependent on $\Delta t$
* equation 12 is wrong, you need to take the gradient of the expected value with respect to the action as well (the dynamics uses $a_t$)
* in the abstract, "A fairly reliable trend in deep reinforcement learning is that performance scales with
the number of parameters, provided a complimentary scaling in amount of training
data. As the appetite for large models increases, it is imperative to address, sooner
than later, the potential problem of running out of high-quality demonstrations." - These statements are not entirely correct, perhaps the authors meant a reliable trend in "supervised learning"? Also it is not clear where demonstrations have come in when the first sentence discusses reinforcement learning.
* miscellaneous typos and notation mistakes in the methods section, interspersed throughout, I pointed the most obvious ones above

In general, I highly recommend the authors re-write the paper for clarity, add proper framing and perspective, improve the methods section considerably and include significantly more comparisons to relevant, SOTA work. In its current form, I do not believe this paper is ready for publication at a venue such as ICLR.

**Questions:**

1. In Figure 5, "Returns are normalized by average per-task data performance." What does this mean precisely?
2. Please evaluate DVF on the full suite of D4RL tasks as done in Diffusion-QL Table 1 so that we can evaluate the complete performance profile of DVF
3. Please provide concrete discussion of DVF differences/tradeoffs relative to other Diffusion-based offline RL methods
4. Why was Perceiver I/O used instead of a standard Diffusion U-Net architecture?
5. Add clarity on which networks are beings trained, their objectives, their inputs and outputs in the methods section. This took a lot of effort to parse from the current methods section.

---

> ### Author Response · Authors · 2023-11-17
> **Author response**
>
> We thank the reviewer for their thorough review of our work, and provide an itemized response below:
>
> > Method clarifications and averaging step
>
> We agree with the reviewer that some additional details can be added to the paper. Specifically, the averaging of $\Delta t$ in the value function estimation is mentioned in Algorithm 1 line 7, which we didn’t mention separately in the main text due to lack of space. The networks architecture and choice is discussed in Appendix Section 7.1, to which we added the architecture of the reward and policy networks as well (4-layer MLPs with mish activation functions, layernorm and residual connections). Figure 5 shows the performance of each method rescaled to the 0-1 interval for each run using the minimum and maximum statistics of the dataset’s logging policy (available on the dataset’s website https://github.com/takuseno/d4rl-pybullet). The mean and standard deviation are then computed on top of these normalized scores.
>
> > Typos and correctness of equations
>
> We appreciate the reviewer’s comments on catching bad phrasing, e.g. “performance of deep reinforcement learning scales with model capacity”, where it should read “performance of offline reinforcement and imitation learning”. Recent works in offline RL [1] and imitation learning [2,3] do show that offline settings (essentially supervised learning as pointed out by the reviewer) do exhibit such trends. Next, Equation 12 indeed has a typo, where the gradient from the dynamics is ignored. We added this term back, per reviewer’s suggestion, and we want to point out that the dependence of dynamics in $a_t$ is implicitly encoded via policy conditioning in $\rho$. Our claim still stands, in that the gradient of $Q$ depends on the gradient of the reward function, which is less computationally cumbersome to compute than if we directly took the gradient of $\epsilon_\theta$ with respect to the action.
>
> > Computational costs of DVF and architecture
>
> The computational costs of DVF primarily depend on the choice of the noise network’s architecture. We investigated the trade-offs between a classical U-net architecture [4], and a cross-attentional model using Perceiver IO [5]. While the U-net is the go-to choice for diffusion in image space, Perceiver worked best in our setting, where policy conditioning requires operating over (potentially) long context windows composed of vectors that vary according to different scales (which is not the case for pixels). We would like to specify that Perceiver IO and U-net are architectural design choices, that are not critical to the overall contribution. The computational cost also depends on the context length which, in our experiments, were fairly short, due to the determinism of the MuJoCo environments. Specifically, we only needed a context of length 4 for all experiments. This makes DVF computationally heavier than expected TD-based approaches, but on-par with diffusion based approaches. One advantage DVF has over other methods is that by decoupling the diffusion model training from the policy training, we can pool together datasets with missing actions or rewards, e.g. train the diffusion model on video data (and using only states as context), and then combine it with a small amount of action-reward labeled data to obtain a more confident statistical estimator of the value function.
>
> > Comparison to other diffusion-based approaches
>
> We agree that our experimental section could benefit from additional baselines that are also based on generative models such as [6,7]. We present the results below, as well as in Table 1 of the revised manuscript.
>
> | **Task**                | **BC** | **CQL** | **IQL** | **BRAC-p** | **BEAR** | **Diffusion BC** | **Diffusion QL** | **DVF**       | **DVF (pooled)** |
> |:-----------------------:|:------:|:-------:|:-------:|:----------:|:--------:|:----------------:|:----------------:|:-------------:|:----------------:|
> | **kitchen-complete-v0** | 65.0   | 43.8    | 62.5    | 0.0        | 0.0      | 53.8 $\pm$ 10    | 67.2 $\pm$ 8     | 74.8 $\pm$ 13 | 79.1 $\pm$ 12    |
> | **kitchen-partial-v0**  | 38.0   | 49.8    | 46.3    | 0.0        | 0.0      | 42.5 $\pm$ 3     | 48.1 $\pm$ 4     | 50.2 $\pm$ 7  | 72.2 $\pm$ 9     |
> | **kitchen-mixed-v0**    | 51.5   | 51.0    | 51.0    | 0.0        | 0.0      | 48.3 $\pm$ 2     | 53.5 $\pm$ 6     | 51.1 $\pm$ 7  | 67.4 $\pm$ 6     |
>
>
> The new baselines based on diffusion roughly match the performance of DVF, while being trained on x3 more compute. Note that we had to re-run the author’s code for [7] in our setting, specifically using a single set of hyperparameters for all runs.
> The main differences of DVF with existing baselines are in the distribution that they model: Diffuser [8] generates action plans using diffusion, Diffusion BC [6] imitates data contained in the training set, and Diffusion QL [9] replaces the SAC [9] policy with a diffusion model.

---

> > ### Author Response · Authors · 2023-11-17
> > **Author response (references)**
> >
> > ## References:
> >
> > [1] Kumar, Aviral, et al. "Offline q-learning on diverse multi-task data both scales and generalizes." arXiv preprint arXiv:2211.15144 (2022).
> >
> > [2] Lee, Kuang-Huei, et al. "Multi-game decision transformers." Advances in Neural Information Processing Systems 35 (2022): 27921-27936.
> >
> > [3] Reed, Scott, et al. "A generalist agent." arXiv preprint arXiv:2205.06175 (2022).
> >
> > [4] Ho, Jonathan, Ajay Jain, and Pieter Abbeel. "Denoising diffusion probabilistic models." Advances in neural information processing systems 33 (2020): 6840-6851.
> >
> > [5] Jaegle, Andrew, et al. "Perceiver io: A general architecture for structured inputs & outputs." arXiv preprint arXiv:2107.14795(2021).
> >
> > [6] Chi, Cheng, et al. "Diffusion policy: Visuomotor policy learning via action diffusion." arXiv preprint arXiv:2303.04137 (2023).
> >
> > [7] Wang, Zhendong, Jonathan J. Hunt, and Mingyuan Zhou. "Diffusion policies as an expressive policy class for offline reinforcement learning." arXiv preprint arXiv:2208.06193(2022).
> >
> > [8] Janner, Michael, et al. "Planning with diffusion for flexible behavior synthesis." arXiv preprint arXiv:2205.09991 (2022).
> >
> > [9] Haarnoja, Tuomas, et al. "Soft actor-critic algorithms and applications." arXiv preprint arXiv:1812.05905 (2018).

---

> > > ### Author Response · Authors · 2023-11-23
> > > **Author response**
> > >
> > > Dear reviewer nnB2,
> > >
> > > We hope we have addressed your questions in our rebuttal. If so - would you consider updating your score?

---

### Official Review · Reviewer_65QG · 2023-11-01

**Soundness:** 3 good
**Presentation:** 3 good
**Contribution:** 3 good
**Rating:** 6
**Confidence:** 3

**Summary:**

This paper proposes a novel method for value function estimation using conditional diffusion models for continuous control tasks. The method learns a generative model of the discounted state occupancy measure from state sequences without reward or action labels, and then uses it to estimate the value function and the optimal action. The paper shows that the method can handle complex robotic tasks, offline reinforcement learning, and exploration from offline data, and outperforms existing baselines.

**Strengths:**

- It proposes a novel algorithm DVF, for value function estimation using diffusion models without requiring reward or action labels.
- It demonstrates that DVF can handle complex robotic tasks and outperforms existing baselines in both online and offline settings.
- It shows how DVF can be used for learning exploration policies from offline datasets, enhancing the efficiency of tabula rasa learning.

**Weaknesses:**

See questions.

**Questions:**

1. The assuption that the behavior policy $\mu$ is known is not usual in offline RL. It seems like this paper only utilized the dataset $\mathcal{D}$. So can this assuption be removed without affecting the result?
1. There has been many works that apply diffusion models on offline RL [1,2,3,etc.] . Could you please include more baselines that use diffusion models for more convincing experiments? Such works are also worth discussing in related works or other parts of the paper.

[1] Diffusion Policies as an Expressive Policy Class for Offline Reinforcement Learning. https://arxiv.org/abs/2208.06193

[2] Is Conditional Generative Modeling all you need for Decision-Making? https://arxiv.org/abs/2211.15657

[3] IDQL: Implicit Q-Learning as an Actor-Critic Method with Diffusion Policies. https://arxiv.org/abs/2304.10573

---

> ### Author Response · Authors · 2023-11-17
> **Author response**
>
> We thank the reviewer for their questions, as well as suggestions of relevant baselines. We provide a detailed response below:
>
> > Knowledge of behavior policy $\mu$
>
> You are correct, in that, generally speaking, the behavior policy that generated the dataset is not known in offline RL. The experiments in Figures 3 and 4, which use the discrete policy representation (i.e. $\phi(\pi_i)=i$ is saved during dataset generation), are provided only for purposes of illustrating the diffusion model’s capabilities to capture long-horizon dependencies and disambiguate data coming from different sources. In practice, we can treat the history of all available data from $\mu$ (or $\mu_i$ if data was collected from a mixture of policies) up to timestep $t$ as the continuous policy representation (or context) of state $s_t$, mimicking the belief state in POMDPs. Such an approach embeds the policy using its trajectories, which is expensive but fairly universal as to forego additional assumptions about the data.
>
> > Additional diffusion baselines
>
> We agree that our experimental section could benefit from additional baselines that are also based on generative models such as [1,2]. We present the results below, as well as in Table 1 of the revised manuscript.
>
> | **Task**                | **BC** | **CQL** | **IQL** | **BRAC-p** | **BEAR** | **Diffusion BC** | **Diffusion QL** | **DVF**       | **DVF (pooled)** |
> |:-----------------------:|:------:|:-------:|:-------:|:----------:|:--------:|:----------------:|:----------------:|:-------------:|:----------------:|
> | **kitchen-complete-v0** | 65.0   | 43.8    | 62.5    | 0.0        | 0.0      | 53.8 $\pm$ 10    | 67.2 $\pm$ 8     | 74.8 $\pm$ 13 | 79.1 $\pm$ 12    |
> | **kitchen-partial-v0**  | 38.0   | 49.8    | 46.3    | 0.0        | 0.0      | 42.5 $\pm$ 3     | 48.1 $\pm$ 4     | 50.2 $\pm$ 7  | 72.2 $\pm$ 9     |
> | **kitchen-mixed-v0**    | 51.5   | 51.0    | 51.0    | 0.0        | 0.0      | 48.3 $\pm$ 2     | 53.5 $\pm$ 6     | 51.1 $\pm$ 7  | 67.4 $\pm$ 6     |
>
>
> The new baselines based on diffusion roughly match the performance of DVF, while being trained on x3 more compute. Note that we had to re-run the author’s code for [2] in our setting, specifically using a single set of hyperparameters for all runs, which the authors do not do in their code.
>
> ## References:
>
> [1] Chi, Cheng, et al. "Diffusion policy: Visuomotor policy learning via action diffusion." arXiv preprint arXiv:2303.04137 (2023).
>
> [2] Wang, Zhendong, Jonathan J. Hunt, and Mingyuan Zhou. "Diffusion policies as an expressive policy class for offline reinforcement learning." arXiv preprint arXiv:2208.06193(2022).

---

> > ### Comment · Reviewer_65QG · 2023-11-21
> >
> > I appreciate for the detailed response from the authors. The addional experiment results have clear my main confusion, so I am glad to raise the rating.

---

### Official Review · Reviewer_zxfx · 2023-11-01

**Soundness:** 3 good
**Presentation:** 3 good
**Contribution:** 3 good
**Rating:** 8
**Confidence:** 3

**Summary:**

This paper proposes a method for learning a value function of a policy by training a generative model of the occupation measure given features of the policy. The authors propose to use state samples from the current policy to train a diffusion model, and weight them by the reward in order to predict the value function. Furthermore, they propose to improve the policy with the estimated value function by taking gradients through the reward. The authors show the efficacy of their algorithm on tasks that are mostly in offline RL.

**Strengths:**

1. The work presents an interesting idea that hasn't been tried by other previous works, the idea of diffusing an occupation measure is quite interesting.
2. The work makes good theoretical connections with existing works in RL and the proposed approach.
3. The incorporation of exploration from data is also quite interesting, as this is rarely considered.

**Weaknesses:**

1. The presentation of the work can be improved as the manuscript is a bit hard to understand in its current form. It would be good to dissect and analyze each sentence with a bit more care when rewriting. I will list some of the points here but these are not isolated issues, I think the authors' work could be presented with much more clarity if written more clearly.
- In Section 2, $\Delta t$ suddenly appears without defining, and the readers are left to figure out what it is.
- The wording of explicit conditioning is also a bit strange in this section, and it requires some domain-expertise to understand what the authors mean by this. The occupation measure is always conditioned on a policy, the choice of whether we implicitly do it or explicitly do it seems like a choice of implementation. Perhaps it's better to say something like "rather than statistically estimating the occupation measure through Monte Carlo sampling, we choose to directly learn a map that can infer the occupation measure given some features of the policy"?
- In Equation 8, $l_{diffusion}$ should explicitly be noted as the function of $\theta$.
- In Section 3, the authors say maximizing $Q(s,a,\phi(\pi))$ directly is expensive, but the readers don't have the context to understand this at this point of the manuscript (we don't yet have the details of what parameters are being maximized, and what is being represented by a diffusion model) and $Q(s,a,\phi(\pi)$ has never been defined anywhere.

2. The baselines in the empirical results are a bit weak as the only compare to BC and CQL. It would have been more informative to include other approaches in offline RL (e.g.Implicit Q-learning, Trajectory Transformer (TT), Diffuser, Score-Guided Planning (SGP)).

**Questions:**

1. The computational aspect of the approach has been relatively not discussed. Is DVF cheaper / more expensive to trained compared to other baselines?
2. Are there other interesting uses cases of having a generative model for the occupation measure besides estimating the value function?

---

> ### Author Response · Authors · 2023-11-17
> **Author response**
>
> We thank the reviewer for their constructive feedback. We provide a point-by-point response below.
>
> > General presentation of the paper
>
> We agree with the reviewer that some parts of the paper warrant clarifications. We made edits to the rebuttal PDF (marked in red), to make the manuscript easier to follow, and correct typos pointed out by the reviewers.
>
> > Missing $\Delta t$
>
> Thanks for catching this, we added the definition back in the footnote of Section 2 page 2.
>
> > Wording of the conditioning operator
>
> We modified the text of Section 3 page 3 to reflect your suggestion, with which we agree. Since our work primarily tackles the challenges arising in offline settings, it is impossible to collect on-policy samples during policy optimization. We’ve experimented with two potential solutions to this issue. The first strategy is to simply condition the occupancy measure (i.e. noise network in diffusion) on data available in the training set. This approach maximizes the value of the behavior policy, and yields a single step of policy improvement which, in most realistic MDPs, is enough to find a policy with higher returns [1,2]. The second strategy is re-generate the policy embedding $\phi(\pi)$ for the learned policy, which can be done with continuous embeddings by using $\rho$ and $\pi$ to predict the future states in an autoregressive manner (see end of Section 3.4). Since the context for deterministic environments like MuJoCo is rather small, this doesn’t pose a significant overhead.
>
> > Equation 8 dependence on $\theta$
>
> Thank you for the observation, we fixed it in the rebuttal PDF Equation 8.
>
> > Computational cost of maximizing $Q(s,a,\phi(\pi))$ directly
>
> The reason for the computational overhead of the direct maximization is due to the fact that the soft policy update (e.g. using SAC’s actor loss[3]) requires to compute $\nabla_{a}Q(s,a,\phi(\pi))$, which in turns needs $\nabla_{a}\epsilon_\theta(s,a,\phi(\pi))$. When $\epsilon_\theta$ is a large network, especially using cross-attention, this computation can be quite memory intensive and FLOP heavy. Instead, we can use a trick similar to [3], where we represent $Q^\pi(s_t,a_t) = r(s_t,a_t)+\gamma E_{s_{t+1}}[V^\pi(s_{t+1})]$, making the gradient only depend on the reward network, which is much smaller. Note that this problem doesn’t arise when the policy is implicitly encoded in the Q-value, e.g. with discrete action spaces. We acknowledge that this point was understated in the paper (Section 3.2), and added a clarifying discussion at the beginning of Section 3.
>
> > Baselines
>
> We agree that our experimental section could benefit from additional baselines that are also based on generative models such as [3,4]. We present the results below, as well as in Table 1 of the revised manuscript.
>
> | **Task**                | **BC** | **CQL** | **IQL** | **BRAC-p** | **BEAR** | **Diffusion BC** | **Diffusion QL** | **DVF**       | **DVF (pooled)** |
> |:-----------------------:|:------:|:-------:|:-------:|:----------:|:--------:|:----------------:|:----------------:|:-------------:|:----------------:|
> | **kitchen-complete-v0** | 65.0   | 43.8    | 62.5    | 0.0        | 0.0      | 53.8 $\pm$ 10    | 67.2 $\pm$ 8     | 74.8 $\pm$ 13 | 79.1 $\pm$ 12    |
> | **kitchen-partial-v0**  | 38.0   | 49.8    | 46.3    | 0.0        | 0.0      | 42.5 $\pm$ 3     | 48.1 $\pm$ 4     | 50.2 $\pm$ 7  | 72.2 $\pm$ 9     |
> | **kitchen-mixed-v0**    | 51.5   | 51.0    | 51.0    | 0.0        | 0.0      | 48.3 $\pm$ 2     | 53.5 $\pm$ 6     | 51.1 $\pm$ 7  | 67.4 $\pm$ 6     |
>
>
> The new baselines based on diffusion roughly match the performance of DVF, while being trained on x3 more compute. Note that we had to re-run the author’s code for [4] in our setting, specifically using a single set of hyperparameters for all runs, which the authors do not do in their code.

---

> > ### Author Response · Authors · 2023-11-17
> > **Author response (continued)**
> >
> > > Computational costs of DVF
> >
> > The computational costs of DVF primarily depend on the choice of the noise network’s architecture. We investigated the trade-offs between a classical U-net architecture [5], and a cross-attentional model using Perceiver IO [6]. While the U-net is the go-to choice for diffusion in image space, Perceiver worked best in our setting, where policy conditioning requires operating over (potentially) long context windows composed of vectors that vary according to different scales (which is not the case for pixels). Perceiver IO also gives a way to scale complexity using the number of latents in addition to the number of layers. The computational cost also depends on the context length which, in our experiments, were fairly short, due to the determinism of the MuJoCo environments. Specifically, we only needed a context of length 4 for all experiments. This makes DVF computationally heavier than expected TD-based approaches, but on-par with diffusion based approaches. One advantage DVF has over other methods is that by decoupling the diffusion model training from the policy training, we can pool together datasets (see last column of Table 1) with missing actions or rewards, e.g. train the diffusion model on video data (and using only states as context), and then combine it with a small amount of action-reward labeled data to obtain a more confident statistical estimator of the value function.
> >
> > > Additional applications of generative modeling of the occupancy measure
> >
> > Thanks for the question - there are indeed a few additional applications, with the most prominent one potentially being using the generative model to estimate state coverage and uncertainty. We took a first step towards this idea in Section 4.4, where we analyzed how well various offline RL methods fare under reward-shaping for exploration. In practice, we ran all methods in Table 2 using some form of reward shaping, and features learned by DVF’s diffusion model were useful for exploration from offline data (i.e. no updates on online samples).
> >
> > ## References:
> >
> > [1] Kakade, Sham, and John Langford. "Approximately optimal approximate reinforcement learning." Proceedings of the Nineteenth International Conference on Machine Learning. 2002.
> >
> > [2] Brandfonbrener, David, et al. "Offline rl without off-policy evaluation." Advances in neural information processing systems 34 (2021): 4933-4946.
> >
> > [3] Chi, Cheng, et al. "Diffusion policy: Visuomotor policy learning via action diffusion." arXiv preprint arXiv:2303.04137 (2023).
> >
> > [4] Wang, Zhendong, Jonathan J. Hunt, and Mingyuan Zhou. "Diffusion policies as an expressive policy class for offline reinforcement learning." arXiv preprint arXiv:2208.06193(2022).
> >
> > [5] Ho, Jonathan, Ajay Jain, and Pieter Abbeel. "Denoising diffusion probabilistic models." Advances in neural information processing systems 33 (2020): 6840-6851.
> >
> > [6] Jaegle, Andrew, et al. "Perceiver io: A general architecture for structured inputs & outputs." arXiv preprint arXiv:2107.14795(2021).

---

> ### Comment · Reviewer_zxfx · 2023-11-23
> **Comment**
>
> I would like to thank the authors for the detailed response - I think the additional baselines and the improvements in presentations are good, I have decided to increase to score to 8.

---

### Author Response · Authors · 2023-11-17
**Common response to reviewers**

We thank all reviewers for their feedback on our work. We understand that the main concerns for the paper is the clarity of its presentation, as well as lack of baselines based on diffusion processes. To address the issue of clarity, we made edits directly in the PDF of the manuscript (highlighted in red), which lead to a clearer version of the paper. For instance, we mention that it is possible to obtain good policies from an offline dataset with a single step of policy improvement [1]. We also added two diffusion based baselines based on [2,3], as suggested by the reviewers. In our experimental setting on Franka Kitchen in D4RL, both baselines achieve performance comparable to DVF, while being trained on x3 the computational budget.

## References:
[1] Brandfonbrener, David, et al. "Offline rl without off-policy evaluation." Advances in neural information processing systems 34 (2021): 4933-4946.

[2] Chi, Cheng, et al. "Diffusion policy: Visuomotor policy learning via action diffusion." arXiv preprint arXiv:2303.04137 (2023).

[3] Wang, Zhendong, Jonathan J. Hunt, and Mingyuan Zhou. "Diffusion policies as an expressive policy class for offline reinforcement learning." arXiv preprint arXiv:2208.06193(2022).

---

### Meta-Review · Area_Chair_ysfR · 2023-12-04

**Metareview:**

This paper introduces a novel method to learn a joint multi-step model of the environment-robot interaction dynamics using a diffusion model for improving efficiency in continuous control.

**Reviewers have reported the following strengths:**

- Proposed idea is novel and interesting;
- Experimental results.

**Reviewers have reported the following weaknesses:**

- Quality of writing;
- Lack of sufficient citations;
- Insufficient variety of experiments.

**SAC note**

Given the disagreement, the SAC read the paper. While the revision/rebuttal helped clarifying some of the writing issues (as acknowledged in reviewer-author discussions), the SAC finds the paper largely unreadable to researchers with expertise in OPE. There are many derivation or algorithmic steps that are vague and unclear; for examples, the paper casually mentions estimating $\rho$ for the policy in several places, which is perhaps more difficult than estimating the value function itself and a proper treatment requires non-standard algorithms, see e.g., https://aaai.org/ojs/index.php/AAAI/article/download/4246/4124. Also $\phi(\pi)$ conditioning requires data to come from a mixture of different policies, yet the setup section and Algorithm 1 are written as if it were the standard setting of a single, inseparable dataset. The normalization step in Algorithm 1 implies that states are vectors in R^d but this is not mentioned in the setup section. Examples like these go on. The reviewers and the ACs generally have concerns about the quality of writing, and despite the clarification and improvement provided during the rebuttal period, the SAC believes that the quality of presentation still has a large room for improvement.

**Justification For Why Not Higher Score:**

The paper retains some issues in the quality of writing that I deem important to not argue for a higher score.

**Justification For Why Not Lower Score:**

N/A

---

### Decision · Program_Chairs · 2024-01-16

Reject